# Utility of Human Papillomavirus Testing for Cervical Cancer Screening in Korea

**DOI:** 10.3390/ijerph17051726

**Published:** 2020-03-06

**Authors:** Mee-seon Kim, Eun Hee Lee, Moon-il Park, Jae Seok Lee, Kisu Kim, Mee Sook Roh, Hyoun Wook Lee

**Affiliations:** 1Department of Pathology, Samsung Changwon Hospital, Sungkyunkwan University School of Medicine, Changwon 51353, Korea; kimm2342@gmail.com (M.-s.K.); dalgaebe@hanmail.net (E.H.L.); moonil.park@samsung.com (M.-i.P.); jaeseok3.lee@samsung.com (J.S.L.); kkspath.kim@samsung.com (K.K.); 2Department of Pathology, Dong-A University College of Medicine, Busan 49201, Korea; msroh@dau.ac.kr

**Keywords:** human papillomavirus, cervical cancer, HPV testing, cervical intraepithelial neoplasia, cytology, HPV epidemiology

## Abstract

(1) Background: Cervical cancer is one of the most common cancers in Korean women. This study was performed to discover the utility of HPV (Human Papillomavirus) testing in screening of cervical lesions and to provide the prevalence of HPV and the genotype distribution in a single center of Korea. (2) Methods: A total of 15,141 women who underwent both HPV testing and cervical cytology were enrolled in this retrospective medical record review study. (3) Results: HPV testing showed higher sensitivity than cytology for the detection of histological high-grade squamous lesions. Furthermore, the sensitivity and specificity of HPV testing varied depending on the method used. The BD Onclarity™ HPV assay had higher sensitivity (90%) than the MyHPV CHIP™ kit (all types of HPV: 82%; high-risk HPV: 76%) for high-grade squamous lesions. A combination of MyHPV CHIP™ and cytology detected 90.9% (30/33) of histological high-grade squamous lesions. A combination of BD Onclarity™ HPV assay and cytology detected 96.55% (84/87) of histological high-grade squamous lesions. In addition, HPV prevalence and genotype distribution were different depending on the HPV testing method used. (4) Conclusion: HPV testing showed higher sensitivity than cytology, but the sensitivity and specificity of HPV testing had variation depending on the method used.

## 1. Introduction

Cervical cancer is one of the most common cancers in Korean women, and in 2019, 2856 new cases were detected and 834 deaths occurred [1]. Since the introduction of the Papanicolaou test for cervical screening, the incidence and mortality rates of cervical cancer have been decreasing year by year in Korea. From 1999 to 2014, the age-standardized incidence and mortality rates decreased from 16.3 to 9.0 per 100,000 women and 2.6 to 2.1 per 100,000 women, respectively, in Korea [2]. In 2019, the age-standardized incidence was 6.9 per 100,000 women and the age-standardized mortality rate was 1.6 per 100,000 women [1]. For this reason, the national health insurance service in Korea has provided the Papanicolaou test free of charge biennially to women aged 20 or above. In 1983 and 1984, HPV16 and HPV18 were isolated from cervical cancer, respectively [3], and various types (high-risk) of HPV (Human Papillomavirus) DNA were found in almost all cervical cancer specimens. Human papillomavirus infection has been established as the major etiology of cervical cancer [4,5]. In recent research, primary high-risk HPV screening showed a more improved sensitivity for the detection of cervical lesions than cytology alone [6,7,8,9,10]. This study was performed to discover the utility of HPV testing in the screening of cervical lesions and to provide the prevalence of HPV and the genotype distribution among a single center of Korea. 

## 2. Materials and Methods 

### 2.1. Study Population

This study was approved by the Institutional Review Board (IRB) of Samsung Changwon Hospital, Changwon, Korea (IRB FILE No. 2020-01-002). This study was a retrospective medical record review study and the IRB waived the need for written informed consent. The medical records of Samsung Changwon Hospital between February 2015 and June 2017 were gathered. A total of 15,141 women (aged 15–97; 43 years (median) and 43.36 years (mean)) who underwent both HPV testing and cervical cytology were enrolled in this study from the health checkup center (*n* = 11,142, aged 23–86; 42 years (median) and 41.5 years (mean)) and the gynecology department (*n* = 3999, aged 15–97; 47 years (median) and 47.5 years (mean)) of Samsung Changwon Hospital. The exclusion criteria included having undergone previous surgical procedures of the cervix, including conization, trachelectomy, or hysterectomy. To determine HPV status, the MyHPV CHIP™ kit (Mygene Co. Seoul, Korea) and the BD Onclarity™ HPV assay on the BD Viper™ LT system (BD Diagnostics, Sparks, MD, USA) were used (Table 1). The MyHPV CHIP™ was performed from February 2015 to February 2016 (*n* = 5324, aged 15–97; 43 years (median) and 43.5 years (mean)), and the BD Onclarity™ HPV assay was performed from March 2016 to June 2017 (*n* = 9817, aged 17–91; 43 years (median) and 43.3 years (mean)). Of these 15,141 women, 295 (1.95%) underwent cervical resection, including conization, trachelectomy, or hysterectomy, until August 2019. 

### 2.2. Cytology

The cytological diagnoses were obtained from pathologic reports using electronic medical records retrospectively. The slides had Papanicolaou staining, and the Papanicolaou-stained slides had been diagnosed using an optical microscope at the department of pathology, Samsung Changwon Hospital. Three cytologists and five pathologists had been involved in the diagnostic process, and 3999 conventional smears and 11,142 liquid-based cytology tests (all, Surepath) had been performed.

### 2.3. Histological Diagnosis

The histological diagnoses were obtained from pathologic reports using electronic medical records retrospectively. All diagnoses had been performed in the department of pathology, Samsung Changwon Hospital. In the 15,141 women, only 195 women had undergone cervical resection in the department of gynecology, Samsung Changwon Hospital. Formalin-fixed paraffin-embedded (FFPE) blocks had been made using the cervical tissues. All tissues had been sectioned with 5-μm-thick microtome and stained by H&E (hematoxylin and eosin) stain. The stained slides had been diagnosed using an optical microscope by three pathologists. 

### 2.4. HPV Detection by MyHPV CHIP™ 

The MyHPV CHIP™ kit is a PCR (Polymerase chain reaction)-based DNA microarray system. It was approved by the Ministry of Food and Drug Safety of the Republic of Korea (Korea Food and Drug Administration, KFDA) in July 2004. All assays were performed according to the manufacturer’s protocol. Cervical cells for HPV testing were obtained using the soft brushes separately from Pap smears. The brushed tissues were supplied to the MyHPV CHIP™ kits and DNA was isolated. Target HPV DNA was amplified by PCR using specific primers with β-globin DNA (internal control). According to the protocol provided by the MyGene Co, Seoul, Korea, DNA isolation and PCR were performed. The PCR products were denatured by heating for 5 min at 95 °C, and then they were mixed with hybridization solution and applied to the DNA chips. The hybridizations were performed at 43 °C for 90 min, and then the products were washed with saline–sodium phosphate–EDTA and air-dried at room temperature. Hybridized HPV DNA signals were detected using a DNA chip scanner (Perkin-Elmer GmbH, Überlingen, Germany). Positive signals revealed a band of 150 base pairs on the gel electrophoresis of the HPV-PCR. The absence of these 150 base pairs on the gel electrophoresis was considered “HPV-negative”. This system detected 11 types of high-risk HPV (types 16, 18, 31, 33, 35, 39, 45, 51, 52, 56, and 58), 8 types of low-risk HPV (types 6, 11, 34, 40, 42, 43, 44, and 54), and other types of HPV (unspecified HPV).

### 2.5. HPV Detection by BD Onclarity™ HPV Assay

The cervical brush diluents in cases of conventional smear and the aliquots of the SurePath sample were used for BD Onclarity HPV testing. The cervical brush diluents did not require pre-handling before HPV testing, but each SurePath sample was vortexed briefly and a 0.5 mL aliquot was supplied to 1.7 mL of a BD HPV LBC diluent. Then, the samples (cervical brush diluent and LBC diluent) were pre-warmed for 30 min at 120 °C on the BD pre-warming heater with positive and negative controls. The pre-warmed specimens were loaded onto the BD Viper LT System after cooling. Then, all of the steps, including extraction and amplification of target DNA, were performed automatically by the BD Viper LT System. Next, the specimens with fluorescent detector probes were transferred to a three-wall, four-channel real-time PCR system for genotyping with the internal controls in each wall. The signals were detected by the integrated VIPER LT reader. The system was used for high-risk types 16, 18, 31, 45, 51, and 52 and other high-risk genotypes by group (P1: 33/58; P2: 56/59/66; P3: 35/39/68).

### 2.6. Statistical Analysis

All analyses were performed using R version 3.6.2., the R Foundation for Statistical Computing, (Auckland, New Zealand). The difference of HPV high-risk prevalence between MyHPV CHIP™ and the BD Onclarity™ HPV assay was calculated by Fisher’s exact test. The difference between the prevalence of high-risk HPV and the prevalence of cytological squamous lesions was calculated by using the Exact McNemar’s test in the package “exact2×2”. The correlation between high-risk HPV infection and cytological diagnosis was calculated using Spearman’s correlation. To compare the proportion of histological high-grade squamous lesions in the negative cytology/positive HPV group with the positive cytology/negative HPV group, Fisher’s exact test was performed. The odds ratio of HPV infection in the normal cytology group and the positive cytology group was calculated by Fisher’s exact test. The sensitivity, specificity, positive predictive value (PPV), negative predictive value (NPV), positive likelihood ratio (PLR), and negative likelihood ratio (NLR) were calculated by using “epi.tests” in the package “epiR”. The differences of sensitivity and specificity in MyHPV CHIP™, the BD Onclarity™ HPV assay, and cervical cytology were calculated by using Exact McNemar’s test in the package “exact2×2”. The ROC (receiver operating characteristic) curves drawn by using ROC in the package ”Epi”.

## 3. Results

### 3.1. The Prevalence of HPV Infection

From February 2015 to February 2016, the MyHPV CHIP™ method was conducted for HPV detection. HPV genotypes in the MyHPV CHIP™ group are shown in Table 2. The HPV results were positive in 737 out of 5324 women (13.84%) in MyHPV CHIP™. In total, of the 737 HPV infections, 427 (57.94%) were high risk, 80 (10.85%) were low risk, and 230 (31.21%) were of other types. Furthermore, 413 (56.04%) of the women had single HPV infection, 77 (10.45%) had double infection, 13 (1.76%) had triple infection, and 4 (0.54%) had quadruple infection. 

From March 2016 to June 2017, the BD Onclarity™ HPV assay was conducted for HPV detection. HPV genotypes in the BD Onclarity™ HPV assay group are shown in Table 3. The HPV tests were positive in 1124 out of 9817 women (11.45%) in the BD Onclarity™ HPV assay. 

In the MyHPV CHIP™ group, 427/5324 (8.02%) had high-risk HPV and 4587/5324 (86.16%) showed negative for HPV. However, in the BD Onclarity™ HPV assay group, 1124/9817 (11.45%) had high-risk HPV and 8693/9817 (88.55%) showed negative for HPV. High-risk HPV was found in significantly higher prevalence in the BD Onclarity™ HPV assay than MyHPV CHIP™ (11.45% vs. 8.02%, *p* = 1.496 × 10^−11^). The prevalence of HPV is shown in Table 4 and Table 5. HPV16 was the most common type (11.5%), and HPV58 was the second most common type (10.09%) in the MyHPV CHIP™ group. However, in the BD Onclarity™ HPV assay, HPV52 was the most common type (17.26%), and HPV51 was the second most common type (9.75%). HPV51 and 52 had a significantly higher prevalence in the BD Onclarity™ HPV assay than in MyHPV CHIP™ (HPV51: 9.75% vs. 1.64%, *p* = 8.376 × 10^−16^; HPV52: 17.26% vs. 5.05%, *p* < 2.2 × 10^−16^). 

The age-related prevalence of high-risk HPV infection is shown in Table 6 and Table 7. In the MyHPV CHIP™ group, those aged 60–69 had the highest prevalence of high-risk HPV (12.8%, 35/274), but in the BD Onclarity™ HPV assay group, those under the age of 30 had the highest prevalence of high-risk HPV (25.8%, 199/771). Under the age of 30, the MyHPV CHIP™ group had a significantly lower prevalence of high-risk HPV than the BD Onclarity™ HPV assay group (12.3% vs. 25.8%; odds ratio 0.4, *p* < 1.044 × 10 ^−8^).

### 3.2. HPV Infection and Cytological Diagnosis

In the 5324 women that underwent MyHPV CHIP™ HPV testing, 427 (8.02%) had high-risk HPV infection, 310 (5.82%) had low-risk or another type of HPV infection, and 4587 (86.16%) showed negative for HPV infection. In the 5315 satisfactory cytology cases, 318 (5.98%) had squamous lesion (ASCUS (Atypical squamous cells of undetermined significance) or worse); 125 cases of the 427 high-risk HPV infection group (29.27%), 56 cases of the 310 low-risk and other types of HPV infection group (18.06%), and 137 cases of the 4578 negative HPV infection group (2.99%) had cytological squamous lesions (ASCUS or worse). There was a difference between the prevalence of high-risk HPV (8.03%, 427/5315) and the prevalence of cytological squamous lesions (5.98%, 318/5315), and the difference was statistically significant (*p* = 1.099 × 10^−6^) (Table 8).

In the 9817 women that underwent BD Onclarity™ HPV assay, 1124 (11.45%) had high-risk HPV infection and 8693 (88.55%) showed negative for HPV infection. In the 9806 satisfactory cytology cases, 576 (5.87%) had squamous lesion (ASCUS or worse); 310 cases of the 1124 high-risk HPV infection group (27.58%) and 266 cases of the 8693 negative HPV infection group (3.06%) had cytological squamous lesions (ASCUS or worse). There was a difference between the prevalence of high-risk HPV (11.46%, 1124/9806) and the prevalence of cytological squamous lesions (5.87%, 576/9806), and the difference was statistically significant (*p* < 2.2 × 10^−6^) (Table 9).

The MyHPV CHIP™ HPV (high-risk) result and the cytology (squamous lesions) showed a significant correlation (rho: 0.280, 95% CI 0.255–0.305; *p* < 2.2 × 10^−16^). Additionally, the BD Onclarity™ HPV assay result and the cytology (squamous lesion) showed a significant correlation (rho 0.325, 95% CI 0.307–0.342; *p* < 2.2 × 10^−16^). 

### 3.3. HPV Infection and Histological Diagnosis

With the MyHPV CHIP™ HPV method, 6.38% of those with negative cytology (threshold: ASCUS or worse)/negative HPV infection had histological high-grade squamous lesions (CIN2 or worse). Furthermore, 30.77% of those with negative cytology (threshold: ASCUS or worse)/positive high-risk HPV infection had high-grade squamous lesions, and 13.04% of those with positive cytology (ASCUS or worse)/negative HPV infection also had high-grade squamous lesions. There was a difference between those with negative cytology/positive high-risk HPV (30.77%, 4/13) and those with positive cytology/negative HPV (13.04%, 3/23) in the prevalence of high-grade cervical lesions, but it was not statistically significant (*p* = 0.225). Additionally, 67.74% of those with positive cytology (ASCUS or worse)/positive high-risk HPV had high-grade squamous lesions in their histologic diagnosis; 5.36% of those with negative cytology (threshold: LSIL or worse)/negative HPV infection had histological high-grade squamous lesions; 27.78% of those with negative cytology (LSIL or worse)/positive high-risk HPV infection and 21.43% of those with positive cytology (LSIL or worse)/negative HPV infection had high-grade squamous lesions; and 76.92% of those with positive cytology (LSIL or worse)/positive high-risk HPV infection had high-grade cervical lesions in their histologic diagnosis. From the total 33 histological high-grade squamous lesion cases, three showed negative cytology and negative HPV with the MyHPV CHIP™ method. This means that the combination of MyHPV CHIP™ and cytology detected 90.9% (30/33) of histological high-grade squamous lesions (Table 10).

With the BD Onclarity™ HPV assay, 6% of those with negative cytology (threshold: ASCUS or worse)/negative HPV infection had high-grade squamous lesions (CIN2 or worse) in their histologic diagnosis; and 56.52% of those with negative cytology (ASCUS or worse)/positive high-risk HPV infection and 33.33% of those with positive cytology (ASCUS or worse)/negative HPV infection had high-grade squamous lesions. There was a difference between negative cytology/positive HPV (56.52%, 13/23) and positive cytology/negative HPV (33.33%, 6/18) in prevalence of high-grade squamous lesions, but it was not statistically significant (*p* = 0.209). Furthermore, 80.25% of those with positive cytology (ASCUS or worse)/positive high-risk HPV infection had high-grade squamous lesions in their histologic diagnosis; 5.45% of those with negative cytology (threshold: LSIL or worse)/negative HPV infection had histological high-grade squamous lesions; 65.79% of those with negative cytology (LSIL or worse)/positive high-risk HPV infection and 46.15% of those with positive cytology (LSIL or worse)/negative HPV infection had high-grade squamous lesions; and 80.3% of those with positive cytology (LSIL or worse)/positive high-risk HPV infection had high-grade squamous lesions in their histologic diagnosis. From the total 87 histological high-grade squamous lesion cases, 3 showed negative cytology and negative HPV with the BD Onclarity™ HPV assay. This means that the combination of BD Onclarity™ HPV assay and cytology detected 96.55% (84/87) of histological high-grade squamous lesions (Table 11).

In the normal cytology group of MyHPV CHIP™, the odds ratio of HPV infection was 5.64 (95% CI 0.82–44.89; *p* = 0.0425) and the odds ratio of high-risk HPV was 6.38 (95% CI 0.91-51.48; *p* = 0.032) for histological high-grade squamous lesions. In the positive cytology group (ASCUS or worse) of MyHPV CHIP™, the odds ratio of HPV infection was 9.23 (95% CI 2.20-56.59; *p* = 0.00046) and the odds ratio of high-risk HPV was 10.42 (95% CI 2.85-45.76; *p* = 7.825 × 10^−5^) for histological high-grade squamous lesions. In the normal cytology group of the BD Onclarity™ HPV assay, the odds ratio of HPV high-risk infection was 19.19 (95% CI 4.26–124.33; *p* = 4.448 × 10^−6^) for histological high-grade squamous lesions. In the positive cytology group (ASCUS or worse) of the BD Onclarity™ HPV assay, the odds ratio of HPV high-risk infection was 7.9 (95% CI 2.33–29.96; *p* = 0.00019) for histological high-grade squamous lesions.

### 3.4. Performance of HPV Test Combinations and Cytology for the Detection of CIN2 or Worse 

The performance of test combinations for the detection of high-grade squamous lesions (CIN2 or worse) is shown in Table 12. The BD Onclarity™ HPV assay had higher sensitivity (90%, 95% CI 81–95) than MyHPV CHIP™ (all types of HPV: 82%, 95% CI 65–93; high-risk HPV: 76%, 95% CI 58–89) or cervical cytology (ASCUS+: 81%, 95% CI 73–87; LSIL+: 70%, 95% CI 61–78) for high-grade squamous lesions. Especially, the BD Onclarity™ HPV assay showed significantly higher sensitivity than cytology (LSIL or worse) for high-grade squamous lesion detection (*p* = 0.00087). For high-grade squamous lesion detection, the BD Onclarity™ HPV assay or cytology (ASCUS or worse/ LSIL or worse) showed the best sensitivity (97%, 95% CI 90–99/97%, 95% CI 90–99) and the lowest negative likelihood ratio (0.06, 95% CI 0.02–0.19/0.06, 95% CI 0.02–0.17). MyHPV CHIP™ high-risk infection had higher specificity (79%, 95% CI 69–87) than the BD Onclarity™ HPV assay (69%, 95% CI 58–79) or cytology (ASCUS+: 63%, 95% CI 56–71; LSIL+: 75%, 95% CI 68–82) for high-grade squamous lesion detection. Especially, MyHPV CHIP™ high-risk infection showed significantly higher specificity than cytology (ASCUS or worse) for high-grade squamous lesion detection (*p* = 0.00599). MyHPV CHIP™ high-risk infection and cytology (LSIL or worse) showed the best specificity (93%, 95% CI 86–98) and the highest positive likelihood ratio (9.09, 95% CI 4.00–20.65) for high-grade squamous lesion detection. The BD Onclarity™ HPV assay and cytology (ASCUS or worse/LSIL or worse) showed the highest positive predictive value (80%, 95% CI 70–88/80%, 95% CI 69–89) compared to the others. MyHPV CHIP™ or cytology (LSIL or worse), MyHPV CHIP™ high-risk or cytology (LSIL or worse), and the BD Onclarity™ assay or cytology (LSIL or worse) showed the highest negative predictive values (95%, 95% CI 85–99; 95%, 95% CI 85–99; 95%, 95% CI 85–99). The addition of MyHPV CHIP™ to cytology increased sensitivity compared to cytology alone (ASCUS or worse: 91% vs. 81%; LSIL or worse: 91% vs. 70%), but it was not statistically significant (ASCUS or worse: *p* = 0.125; LSIL or worse: *p* = 0.0625). The addition of BD Onclarity™ HPV assay to cytology increased sensitivity compared to cytology alone (ASCUS or worse: 97% vs. 81%; LSIL or worse: 97% vs. 70%), and it was statistically significant (ASCUS or worse: *p* = 0.000244; LSIL or worse: *p* = 5.96 × 10^−8^). The addition of MyHPV CHIP™ to cytology increased specificity compared to cytology alone (ASCUS+/MyHPV CHIP™ all types 82% vs. ASCUS+ 63%; ASCUS+/MyHPV CHIP™ high-risk 89% vs. ASCUS+ 63%; LSIL+/MyHPV CHIP™ all types 87% vs. LSIL + 75%; LSIL + /MyHPV CHIP™ high-risk 93% vs. LSIL + 75%), and it was statistically significant (ASCUS+ vs. ASCUS+ / MyHPV CHIP™ high-risk, *p* = 2.98 × 10^−8^; ASCUS+ vs. ASCUS+/MyHPV CHIP™ all types, *p* = 1.907 × 10^−6^; LSIL+ vs. LSIL+/MyHPV CHIP™ high-risk, *p* = 1.526 × 10^−5^; LSIL+ vs. LSIL+/MyHPV CHIP™ all types, *p* = 0.000977). The addition of BD Onclarity™ HPV assay to cytology increased specificity compared to cytology alone (ASCUS+/BD Onclarity™ HPV 81% vs. ASCUS+ 63%; LSIL+/BD Onclarity™ HPV 85% vs. LSIL+ 75%), and it was statistically significant (ASCUS+ vs. ASCUS+/BD Onclarity™, *p* = 0.00049; LSIL+ vs. LSIL+/BD Onclarity™, *p* = 0.0156). The addition of cytology to MyHPV CHIP™ increased sensitivity compared to MyHPV CHIP™ alone (MyHPV CHIP™ high-risk/ASCUS+ 91% vs. MyHPV CHIP™ high-risk 76%), but it was not statistically significant (*p* = 0.0625). The addition of cytology to BD Onclarity™ HPV assay increased sensitivity compared to BD Onclarity™ HPV assay alone (BD Onclarity™/ASCUS+ 97% vs. BD Onclarity™ 90%), and it was statistically significant (*p* = 0.000488). The addition of cytology to MyHPV CHIP™ increased specificity compared to MyHPV CHIP™ alone (MyHPV CHIP™ high-risk/ASCUS+ 89% vs. MyHPV CHIP™ high-risk 79%), and it was statistically significant (*p* = 0.0039). The addition of cytology to BD Onclarity™ HPV assay increased specificity compared to BD Onclarity™ HPV assay alone (BD Onclarity™/ASCUS+ 81% vs. BD Onclarity™ 69%), and it was statistically significant (p = 0.00195). The addition of HPV testing to cytology increased sensitivity and specificity compared to cytology alone. Also, the addition of cytology to HPV testing increased sensitivity and specificity compared to HPV testing alone. According to these results, HPV testing and cytology were complementary. The ROC curves are shown Figure 1 and Figure 2. 

The performance of the test combinations for the detection of high-grade squamous lesions in those under the age of 45 and the those aged 45 or above is shown in Table 13 and Table 14. Generally, the results of the performance in the two age groups and the result of all age groups were similar. However, in the those under the age of 45, the addition of cytology to BD Onclarity™ HPV assay did not increase sensitivity compared to BD Onclarity™ HPV assay alone.

## 4. Discussion

Traditionally, cervical cancer screening is performed using cytology. Cytology-based screening has reduced the incidence and mortality of cervical cancer in many countries [11]; however, the sensitivity of cytology-based screening is insufficient to prevent cervical cancer [12,13]. According to a meta-analysis, the global prevalence of HPV infection in normal cytology is around 11%–12% [14]. Furthermore, human papillomavirus is known as major etiology of cervical cancer [4]; indeed, persistent high-risk HPV infection induces cervical cancer [15,16]. In recent research from various countries, primary high-risk HPV screening showed an improved sensitivity for the detection of high-grade squamous lesions compared to cytology alone [5,6,7,8,9]. Additionally, HPV testing has been proposed as another method of cervical screening [17]. This study was performed to discover the utility of HPV testing in the screening of high-grade squamous lesions and to provide the prevalence of HPV and the genotype distribution in Korea. 

According to the recent literature, HPV52 and HPV58 present higher prevalence in Asian women than in Western women [18,19,20]. In our study, HPV58 was the second most common type in the MyHPV CHIP™ group and HPV52 was the most common type in the BD Onclarity™ HPV assay group. Similar to the prevalence in Asia, HPV52 and HPV58 present high prevalence in our institution. However, HPV prevalence and genotype distribution were different depending on the HPV testing method used. HPV51 and 52 had significant higher prevalence in the BD Onclarity™ HPV assay group than in the MyHPV CHIP™ group (51: 9.75% vs. 1.64%, *p* = 8.376 × 10^−16^; 52: 17.26% vs. 5.05%, *p* < 2.2 × 10^−16^). According to our results, it is possible that HPV prevalence appears differently depending on the method and the timing of the investigation. Furthermore, high-risk HPV showed a significantly higher prevalence in the BD Onclarity™ HPV assay group than in the MyHPV CHIP™ group (11.45% vs. 8.02%, *p* = 1.496 × 10^−11^). According to a previous meta-analysis, age-related HPV prevalence in countries shows a bimodal curve with a first peak at younger ages, a plateau at middle ages, and a rebound at older ages [21]. In our study, age-related HPV prevalence showed a first peak at under the age of 30, a plateau at aged 30–59, and a rebound at aged 60 or above. This result is similar to the previous meta-analysis. However, there was a difference in prevalence between the two HPV methods in the young age group of our institution. Under the age of 30, MyHPV CHIP™ showed a significantly lower prevalence of high-risk HPV than the BD Onclarity™ HPV assay (12.3% vs. 25.8%, *p* < 1.044 × 10^−8^).

The current studies indicate that HPV testing showed higher sensitivity than cytology, but the sensitivity and specificity of HPV testing varied depending on the HPV testing method used [22,23,24]. Our study also shows similar results to previous studies. The HPV testing showed higher sensitivity than cytology (BD Onclarity™ HPV assay 90%; MyHPV CHIP™ 82%; ASCUS or worse 81%), but there was a slight difference between MyHPV CHIP™ and cytology (threshold: ASCUS or worse). In our study, the BD Onclarity™ HPV assay had higher sensitivity (90%) than the MyHPV CHIP™ kit (all types of HPV: 82%, high-risk HPV: 76%) for high-grade squamous lesions, and MyHPV CHIP™ had higher specificity (79%) than the BD Onclarity™ HPV assay (69%). 

The addition of HPV testing to cytology increased sensitivity and specificity compared to cytology alone. Additionally, the addition of cytology to HPV testing increased sensitivity and specificity compared to HPV testing alone. Thus, HPV testing and cytology were complementary. 

The obvious limitation in this study was a small sample size of confirmed histological diagnoses. Only about 14.5% of women of abnormal cytology or high-risk HPV infection underwent cervical resection in our hospital (19.4% in MyHPV CHIP™; 12.3% in the BD Onclarity™ HPV assay). The majority of cases (73.59%) were women identified through systemic health screening. Many of them underwent cervical resection in other gynecological hospitals and we could not access their histologically confirmed diagnoses. Besides this, we failed to exclude pregnant women and vaccinated women due to their incomplete medical information.

## 5. Conclusions

HPV testing showed higher sensitivity than cytology, and the sensitivity and specificity of HPV testing varied depending on the HPV testing method. HPV prevalence and genotype distribution were also different depending on the method. The BD Onclarity™ HPV assay had higher sensitivity than the MyHPV CHIP™ kit for high-grade squamous lesions. Furthermore, the combination of cytology and HPV testing increased sensitivity and specificity compared to one method alone; thus, cytology and HPV testing are complementary.

## Figures and Tables

**Figure 1 ijerph-17-01726-f001:**
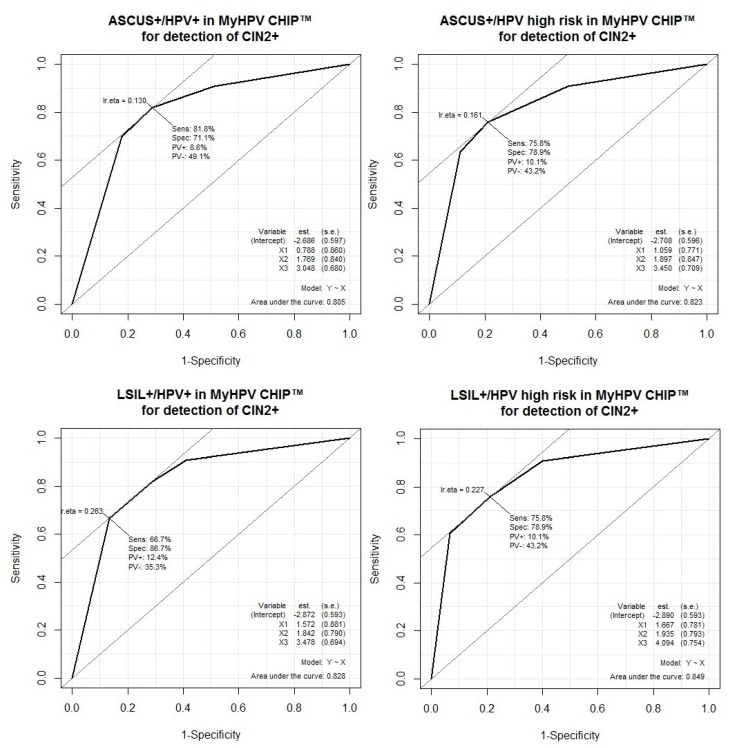
ROC (receiver operating characteristic) curve of cytology and MyHPV CHIP™ for the detection of CIN2 or worse: The AUC (area under the curve) of Cytology (ASCUS or worse)/MyHPV CHIP™ all types is 0.805. The AUC of Cytology (ASCUS or worse)/MyHPV CHIP™ high-risk is 0.823. The AUC of Cytology (LSIL or worse)/MyHPV CHIP™ all types is 0.828. The AUC of Cytology (LSIL or worse)/MyHPV CHIP™ high-risk is 0.849.

**Figure 2 ijerph-17-01726-f002:**
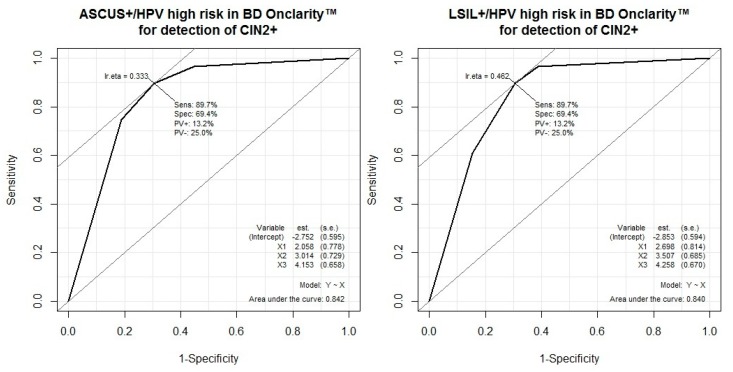
ROC curve of cytology and BD Onclarity™ HPV assay for the detection of CIN2 or worse: The AUC of Cytology (ASCUS or worse)/BD Onclarity™ HPV assay is 0.842. The AUC of Cytology (LSIL or worse)/BD Onclarity™ HPV assay is 0.840.

**Table 1 ijerph-17-01726-t001:** Methods of HPV (Human Papillomavirus) testing.

HPV Test	Method	Genotypes
MyHPV CHIP™	PCR and chip hybridization	11 high-risk HPV: 16, 18, 31, 33, 35, 39, 45, 51, 52, 56, and 58 8 low-risk HPV: 6, 11, 34, 40, 42, 43, 44, and 54
		Other types of HPV: unspecified HPV
BD Onclarity™ HPV Assay	Real time PCR	14 high-risk HPV: 16, 18, 31, 45, 51, 52, P1 (33/58), P2 (56/59/66), and P3 (35/39/68)

PCR: Polymerase chain reaction.

**Table 2 ijerph-17-01726-t002:** HPV Genotypes in the MyHPV CHIP™ group.

	Genotypes	No. (%)		Genotypes	No. (%)
**Sigle infection**	6 *	16 (2.17)	**Double infection**	6, 11 *	2 (0.27)
	11 *	4 (0.54)		6, 16	2 (0.27)
	16	75 (10.18)		6, 18	1 (0.135)
	18	30 (4.07)		6, 39	1 (0.135)
	31	22 (2.99)		6, 42 *	1 (0.135)
	33	20 (2.71)		6, 56	1 (0.135)
	34 *	1 (0.135)		11, 16	1 (0.135)
	35	9 (1.22)		16, 18	3 (0.407)
	39	26 (3.53)		16, 31	1 (0.135)
	40 *	8 (1.09)		16, 33	1 (0.135)
	42 *	9 (1.22)		16, 40	2 (0.27)
	43 *	5 (0.68)		16, 54	2 (0.27)
	44 *	2 (0.27)		16, 58	3 (0.407)
	45	8 (1.09)		18, 33	1 (0.135)
	51	10 (1.35)		18, 44	1 (0.135)
	52	36 (4.88)		18, 45	1 (0.135)
	54 *	30 (4.07)		18, 52	1 (0.135)
	56	43 (5.83)		18, 54	2 (0.27)
	58	59 (8.00)		31, 44	1 (0.135)
	Total	413 (56.04)		31, 54	1 (0.135)
Triple infection	6, 16, 35	1 (0.135)		31, 58	1 (0.135)
	6, 42, 58	1 (0.135)		33, 35	8 (1.09)
	6, 51, 52	1 (0.135)		33, 40	1 (0.135)
	11, 33, 52	1 (0.135)		33, 54	1 (0.135)
	16, 33, 54	1 (0.135)		33, 58	9 (1.22)
	16, 33, 58	1 (0.135)		34, 52	1 (0.135)
	16, 52, 56	1 (0.135)		34, 58	1 (0.135)
	18, 33, 42	1 (0.135)		35, 45	1 (0.135)
	33, 35, 39	1 (0.135)		39, 54	1 (0.135)
	33, 35, 52	1 (0.135)		39, 56	3 (0.407)
	33, 35, 56	1 (0.135)		39, 58	2 (0.27)
	35, 58, 51	1 (0.135)		40, 45	1 (0.135)
	43, 45, 56	1 (0.135)		42, 43 *	1 (0.135)
	Total	13 (1.76)		42, 56	1 (0.135)
Quadruple	16, 11, 33, 58	1 (0.135)		43, 45	3 (0.407)
infection	16, 18, 33, 35	1 (0.135)		43, 52	1 (0.135)
	16, 33, 40, 58	1 (0.135)		43, 54 *	1 (0.135)
	16, 58, 40, 44	1 (0.135)		43, 58	1 (0.135)
	Total	4 (0.54)		45, 56	1 (0.135)
HPV-positive	High-risk	427 (57.94)		51, 54	1 (0.135)
	Low-risk	80 (10.85)		51, 58	1 (0.135)
	Other	230 (31.21)		54, 56	4 (0.54)
	Total	737 (13.84)		54, 58	2 (0.27)
HPV-negative		4587 (86.16)		56, 58	1 (0.135)
Total women		5324		Total	77 (10.45)

* Low-risk HPV.

**Table 3 ijerph-17-01726-t003:** HPV genotypes in the BD Onclarity™ HPV assay group.

Genotypes	No. (%)	Genotypes	No. (%)
16	90 (8.0)	P1 (33/58), P2 (56/59/66)	10 (0.89)
18	20 (1.78)	P1 (33/58), P3 (35/39/68)	7 (0.623)
31	28 (2.49)	P2 (56/59/66), P3 (35/39/68)	14 (1.25)
45	19 (1.69)	16, 18, P2 (56/59/66)	1 (0.089)
51	89 (7.92)	16, 18, P3 (35/39/68)	1 (0.089)
52	165 (14.68)	16, 31, P2 (56/59/66)	1 (0.089)
P1 (33/58)	125 (11.12)	16, 45, P1 (33/58)	1 (0.089)
P2 (56/59/66)	163 (14.5)	16, 45, P3 (35/39/68)	1 (0.089)
P3 (35/39/68)	249 (22.15)	16, 52, P3 (35/39/68)	1 (0.089)
16, 18	1 (0.089)	18, 31, P1 (33/58)	2 (0.178)
16, 31	1 (0.089)	18, 52, P1 (33/58)	1 (0.089)
16, 51	3 (0.267)	18, 51, P2 (56/59/66)	1 (0.089)
16, 52	6 (0.534)	45, 51, P3 (35/39/68)	1 (0.089)
18, 51	4 (0.356)	51, 52, P1 (33/58)	2 (0.178)
18, 52	5 (0.445)	51, 52, P3 (35/39/68)	1 (0.089)
31, 45	1 (0.089)	16, P1 (33/58), P3 (35/39/68)	1 (0.089)
31, 51	1 (0.089)	16, P2 (56/59/66), P3 (35/39/68)	1 (0.089)
31, 52	2 (0.178)	18, P2 (56/59/66), P3 (35/39/68)	1 (0.089)
45, 51	1 (0.089)	31, P2 (56/59/66), P3 (35/39/68)	1 (0.089)
45, 52	1 (0.089)	45, P2 (56/59/66), P3 (35/39/68)	2 (0.178)
51, 52	6 (0.534)	51, P1 (33/58), P3 (35/39/68)	1 (0.089)
16, P1 (33/58)	4 (0.356)	52, P1 (33/58), P3 (35/39/68)	2 (0.178)
16, P2 (56/59/66)	6 (0.534)	51, P1 (33/58), P2 (56/59/66)	1 (0.089)
16, P3 (35/39/68)	3 (0.267)	52, P1 (33/58), P2 (56/59/66)	1 (0.089)
18, P2 (56/59/66)	4 (0.356)	51, P2 (56/59/66), P3 (35/39/68)	1 (0.089)
31, P1 (33/58)	1 (0.089)	52, P2 (56/59/66), P3 (35/39/68)	2 (0.178)
31, P2 (56/59/66)	3 (0.267)	P1 (33/58), P2 (56/59/66), P3 (35/39/68)	3 (0.267)
31, P3 (35/39/68)	2 (0.178)	16, 18, 45, 51	1 (0.089)
45, P1 (33/58)	1 (0.089)	31, 51, 52, P3(35/39/68)	1 (0.089)
45, P3 (35/39/68)	2 (0.178)	16, 18, P1 (33/58), P3 (35/39/68)	1 (0.089)
51, P1 (33/58)	3 (0.267)	45, 52, P2 (56/59/66), P3 (35/39/68)	1 (0.089)
51, P2 (56/59/66)	3 (0.267)	51, 52, P2 (56/59/66), P3 (35/39/68)	1 (0.089)
51, P3 (35/39/68)	10 (0.89)	16, 45, 31, P2 (56/59/66), P3 (35/39/68)	1 (0.089)
52, P1 (33/58)	7 (0.623)	HPV-positive	1124 (11.45)
52, P2 (56/59/66)	13 (1.16)	HPV-negative	8693 (88.55)
52, P3 (35/39/68)	14 (1.25)	Total women	9817

**Table 4 ijerph-17-01726-t004:** HPV genotype prevalence in the MyHPV CHIP™ group.

Genotype	No. of Detection (%)
6 *	27 (3.17)
11 *	9 (1.06)
16	98 (11.5)
18	42 (4.93)
31	26 (3.05)
33	51 (5.99)
34 *	3 (0.35)
35	24 (2.82)
39	34 (3.99)
40 *	14 (1.64)
42 *	14 (1.64)
43 *	13 (1.53)
44 *	5 (0.59)
45	16 (1.88)
51	14 (1.64)
52	43 (5.05)
54 *	46 (5.40)
56	57 (6.69)
58	86 (10.09)
Other	230 (26.99)
Total	852

* low-risk HPV.

**Table 5 ijerph-17-01726-t005:** HPV genotype prevalence in the BD Onclarity™ HPV assay group.

Genotype	No. of Detection (%)
16	125 (9.3)
18	43 (3.2)
31	45 (3.35)
45	33 (2.45)
51	131 (9.75)
52	232 (17.26)
P1 (33/58)	174 (12.95)
P2 (56/59/66)	235 (17.48)
P3 (35/39/68)	326 (24.26)
Total	1344

**Table 6 ijerph-17-01726-t006:** Age-related prevalence of HPV in the MyHPV CHIP™ group.

Age	High-Risk HPV (%)	Total
<30	56 (12.3)	454
30–39	109 (6.9)	1574
40–49	124 (6.9)	1799
50–59	85 (7.9)	1081
60–69	35 (12.8)	274
≥70	18 (12.7)	142
total	427 (8.0)	5324

**Table 7 ijerph-17-01726-t007:** Age related prevalence of HPV in the BD Onclarity™ HPV assay group.

Age	High-Risk HPV (%)	Total
<30	199 (25.8)	771
30–39	330 (11.0)	2988
40–49	290 (8.3)	3489
50–59	201 (11.1)	1808
60–69	69 (12.4)	556
≥70	35 (17.1)	205
total	1124 (11.4)	9817

**Table 8 ijerph-17-01726-t008:** HPV infection and cytologic diagnosis in the MyHPV CHIP™ group.

		Unsatisfactory	NILM	Squamous Lesions					Glandular Lesions			Other Malignancies	Total
				ASCUS	ASCH	LSIL	HSIL	SCC	AGC	AIS	ADC		
HPV-positive	LBC	0	223	31	10	46	25	6	0	0	0	0	341
	Smear	0	78	3	1	1	2	0	1	0	0	0	86
(high-risk)	Total (%)	0 (0)	301 (70.49)	125 (29.27)					1 (0.23)			0 (0)	427 (8.02)
HPV-positive	LBC	0	183	23	1	23	5	0	1	0	0	0	236
	Smear	0	70	3	0	1	0	0	0	0	0	0	74
(low-risk or other)	Total (%)	0 (0)	253 (81.61)	56 (18.06)					1 (0.32)			0 (0)	310 (5.82)
HPV-negative	LBC	7	3049	85	10	23	7	6	7	0	2	2	3198
	Smear	2	1379	4	1	0	1	0	2	0	0	0	1389
	Total (%)	9 (0.196)	4428 (96.53)	137 (2.99)					11 (0.24)			2 (0.044)	4587 (86.16)
Total(%)		9 (0.169)	4982 (93.58)	318 (5.97)					13 (0.244)			2 (0.037)	5324 (100)

NILM: Negative for intraepithelial lesion or malignancy; ASCUS: Atypical squamous cells of undetermined significance; ASCH: Atypical squamous cells- cannot exclude HSIL; LSIL: Low grade squamous intraepithelial lesion; HSIL: High grade squamous intraepithelial lesion; SCC: Squamous cell carcinoma; AGC: Atypical glandular cells; AIS: Adenocarcinoma in situ; ADC: Adenocarcinoma.

**Table 9 ijerph-17-01726-t009:** HPV infection and cytologic diagnosis in the BD Onclarity™ HPV assay group.

		Unsatisfactory	NILM	Squamous Lesions					Glandular Lesions			Other Malignancies	Total
				ASCUS	ASCH	LSIL	HSIL	SCC	AGC	AIS	ADC		
HPV-positive	LBC	0	636	76	31	130	40	10	8	0	0	1	932
	Smear	0	168	15	1	4	3	0	1	0	0	0	192
	Total (%)	0 (0)	804 (71.53)	310 (27.58)					9 (0.80)			1 (0.09)	1124 (11.45)
HPV-negative	LBC	11	6152	189	4	62	4	1	10	0	0	2	6435
	Smear	0	2252	5	1	0	0	0	0	0	0	0	2258
	Total (%)	11 (0.127)	8404 (96.675)	266 (3.06)					10 (0.115)			2 (0.023)	8693 (88.55)
Total (%)		11 (0.112)	9208 (93.796)	576 (5.867)					19 (0.193)			3 (0.031)	9817 (100)

**Table 10 ijerph-17-01726-t010:** Histological diagnosis in the MyHPV CHIP™ group.

**Cytology (threshold:ASCUS +)**	**HPV Infection (MyHPV CHIP™)**		**Negative**	**CIN1**	**CIN2+**	**Other Tumors**	**Endocervical Adenocarcinoma**	**Total**
**Cytology (–)**	**HPV (–)**		39	3	3	2	0	47
			(82.98)	(6.38)	(6.38)	(4.26)	(0)	(38.21)
	HPV(+)	High-risk (%)	0	9	4	0	0	13
			(0)	(69.23)	(30.77)	(0)	(0)	(10.57)
		Low-risk/	0	1	0	0	0	1
		Other types (%)	(0)	(100)	(0)	(0)	(0)	(0.81)
Cytology (+)	HPV (–)		8	8	3	2	2	23
			(34.78)	(34.78)	(13.04)	(8.7)	(8.7)	(18.7)
	HPV (+)	High-risk (%)	1	9	21	0	0	31 (25.2)
			(3.23)	(29.03)	(67.74)	(0)	(0)	
		Low-risk/	0	6	2	0	0	8
		Other types (%)	(0)	(75)	(25)	(0)	(0)	(6.5)
Total (%)			48	36	33	4	2	123
			(39.02)	(29.27)	(26.83)	(3.25)	(1.63)	(100)
**Cytology (threshold:LSIL +)**	**HPV infection (MyHPV CHIP™)**		**Negative**	**CIN1**	**CIN2+**	**Other tumors**	**Endocervical adenocarcinoma**	**Total**
Cytology (–)	HPV (–)		46	5	3	2	0	56
			(82.14)	(8.93)	(5.36)	(3.57)	(0)	(45.53)
	HPV (+)	High-risk (%)	0	13	5	0	0	18
			(0)	(72.22)	(27.78)	(0)	(0)	(14.63)
		Low-risk/	0	1	0	0	0	1
		Other types (%)	(0)	(100)	0	(0)	(0)	(0.81)
Cytology (+)	HPV (–)		1	6	3	2	2	14
			(7.14)	(42.86)	(21.43)	(14.285)	(14.285)	(11.38)
	HPV (+)	High-risk (%)	1	5	20	0	0	26
			(3.85)	(19.23)	(76.92)	(0)	(0)	(21.14)
		Low-risk/	0	6	2	0	0	8
		Other types (%)	(0)	(75)	(25)	(0)	(0)	(6.5)
Total (%)			48	36	33	4	2	123
			(39.02)	(29.27)	(26.83)	(3.25)	(1.63)	(100)

Note: ASCUS +, ASCUS or worse; LSIL +, LSIL or worse; CIN2 +, CIN2 or worse.

**Table 11 ijerph-17-01726-t011:** Histological diagnosis in the BD Onclarity™ HPV assay group.

**Cytology** **(threshold:** **ASCUS +)**	**HPV Infection** **(BD Onclarity™)**	**Negative**	**CIN1**	**CIN2+**	**Other** **Tumors**	**Endocervical** **Adenocarcinoma**	**Total**
Cytology (–) (%)	HPV (–)	42	5	3	0	0	50
		(84)	(10)	(6)	(0)	(0)	(29.07)
	HPV (+)	3	7	13	0	0	23
		(13.04)	(30.43)	(56.52)	(0)	(0)	(13.37)
Cytology (+) (%)	HPV (–)	3	8	6	1	0	18
		(16.67)	(44.44)	(33.33)	(5.56)	(0)	(10.47)
	HPV (+)	2	14	65	0	0	81
		(2.47)	(17.28)	(80.25)	(0)	(0)	(47.09)
Total (%)		50	34	87	1	0	172
		(29.07)	(19.77)	(50.58)	(0.58)	(0)	(100)
**Cytology (threshold: ASCUS +)**	**HPV Infection** **(BD Onclarity™)**	**Negative**	**CIN1**	**CIN2+**	**Other tumors**	**Endocervical Adenocarcinoma**	**Total**
Cytology (–) (%)	HPV (–)	45	7	3	0	0	55
		(81.82)	(12.73)	(5.45)	(0)	(0)	(31.98)
	HPV (+)	5	8	25	0	0	38
		(13.16)	(21.05)	(65.79)	(0)	(0)	(22.09)
Cytology (+) (%)	HPV (–)	0	6	6	1	0	13
		(0)	(46.15)	(46.15)	(7.7)	(0)	(7.56)
	HPV (+)	0	13	53	0	0	66
		(0)	(19.7)	(80.3)	(0)	(0)	(38.37)
Total (%)	Total	50	34	87	1	0	172
		(29.07)	(19.77)	(50.58)	(0.58)	(0)	(100)

Note: ASCUS +, ASCUS or worse; LSIL +, LSIL or worse; CIN2 +, CIN2 or worse.

**Table 12 ijerph-17-01726-t012:** Performance of HPV test combinations and cytology for the detection of CIN2 or worse.

All women	Sensitivity% (95% CI)	Specificity% (95% CI)	PPV% (95% CI)	NPV% (95% CI)	PLR (95% CI)	NLR (95% CI)
HPV+, all types (MyHPV CHIP™)	82 (65–93)	71 (61–80)	51 (37–65)	91 (82–97)	2.83 (1.97–4.07)	0.26 (0.12, 0.53)
HPV+, High-risk (MyHPV CHIP™)	76 (58–89)	79 (69–87)	57 (41–72)	90 (81–96)	3.59 (2.30–5.59)	0.31 (0.17, 0.57)
HPV+, High-risk (BD Onclarity™)	90 (81–95)	69 (58–79)	75 (66–83)	87 (76–94)	2.93 (2.11–4.07)	0.15 (0.08, 0.28)
ASCUS or worse (ASCUS+)	81 (73-87)	63 (56–71)	60 (52–68)	83 (75–89)	2.21 (1.79–2.74)	0.30 (0.21, 0.44)
LSIL or worse (LSIL+)	70 (61–78)	75 (68–82)	66 (57–74)	79 (72–85)	2.85 (2.14–3.79)	0.40 (0.30–0.53)
HPV+ all types (MyHPV CHIP™) or ASCUS+	91 (76–98)	49 (38–60)	39 (28–51)	94 (82–99)	1.78 (1.41–2.24)	0.19 (0.06–0.56)
HPV+, High-risk (MyHPV CHIP™) or ASCUS+	91 (76–98)	50 (39–61)	40 (29–52)	94 (83–99)	1.82 (1.44–2.30)	0.18 (0.06–0.55)
HPV+, Hig- risk (BD Onclarity™) or ASCUS+	97 (90–99)	55 (44–66)	69 (60–77)	94 (83–99)	2.16 (1.70–2.74)	0.06 (0.02–0.19)
HPV+ all types (MyHPV CHIP™) or LSIL+	91 (76–98)	59 (48–69)	45 (33–57)	95 (85–99)	2.21 (1.69–2.90)	0.15 (0.05–0.46)
HPV+, High-risk (MyHPV CHIP™) or LSIL+	91 (76–98)	60 (49–70)	45 (33–58)	95 (85–99)	2.27 (1.73–2.99)	0.15 (0.05–0.45)
HPV+, High-risk (BD Onclarity™) or LSIL+	97 (90–99)	61 (50–72)	72 (63–80)	95 (85–99)	2.49 (1.90–3.26)	0.06 (0.02–0.17)
HPV+ all types (MyHPV CHIP™) and ASCUS+	70 (51–84)	82 (73–89)	59 (42–74)	88 (79–94)	3.92 (2.38–6.45)	0.37 (0.22–0.62)
HPV+, High-risk (MyHPV CHIP™) and ASCUS+	64 (45–80)	89 (81–95)	68 (49–83)	87 (78–93)	5.73 (3.02–10.85)	0.41 (0.26–0.65)
HPV+, High-risk (BD Onclarity™) and ASCUS+	75 (64–83)	81 (71–89)	80 (70–88)	76 (66–84)	3.97 (2.51–6.28)	0.31 (0.21–0.45)
HPV+ all types (MyHPV CHIP™) and LSIL+	67 (48–82)	87 (78–93)	65 (46–80)	88 (79–94)	5.00 (2.80–8.92)	0.38 (0.24–0.63)
HPV+, High-risk (MyHPV CHIP™) and LSIL+	61 (42–77)	93 (86–98)	77 (56–91)	87 (78–93)	9.09 (4.00–20.65)	0.42 (0.28–0.65)
HPV+, High-risk (BD Onclarity™) and LSIL+	61 (50–71)	85 (75–92)	80 (69–89)	68 (58–77)	3.98 (2.35–6.75)	0.46 (0.35–0.61)

**Table 13 ijerph-17-01726-t013:** Performance of HPV test combinations and cytology for the detection of CIN2 or worse (Age <45).

Age < 45	Sensitivity% (95% CI)	Specificity% (95% CI)	PPV% (95% CI)	NPV% (95% CI)	PLR (95% CI)	NLR (95% CI)
HPV+, all types (MyHPV CHIP™)	89 (52–100)	58 (37–78)	44 (22–69)	93 (68–100)	2.13 (1.26–3.61)	0.19 (0.03–1.25)
HPV+, High-risk (MyHPV CHIP™)	67 (30–93)	75 (53–90)	50 (21–79)	86 (64–97)	2.67 (1.16–6.13)	0.44 (0.17–1.15)
HPV+, High-risk (BD Onclarity™)	97 (83–100)	54 (25–81)	83 (66–93)	88 (47–100)	2.09 (1.16–3.78)	0.06 (0.01–0.45)
ASCUS or worse (ASCUS+)	77 (61–89)	70 (53–84)	73 (57–.86)	74 (57–88)	2.59 (1.53–4.37)	0.33 (0.18–0.60)
LSIL or worse (LSIL+)	67 (50–81)	81 (65–92)	79 (61–91)	70 (54–.83)	3.52 (1.74–7.12)	0.41 (0.26–0.66)
HPV+ all types (MyHPV CHIP™) or ASCUS+	89 (52–100)	46 (26–67)	38 (18–62)	92 (62–100)	1.64 (1.06–2.53)	0.24 (0.04–1.62)
HPV+, High-risk (MyHPVCHIP™) or ASCUS+	89 (52–100)	50 (29–71)	40 (19–64)	92 (64–100)	1.78 (1.12–2.82)	0.22 (0.03–1.47)
HPV+, High-risk (BD Onclarity™) or ASCUS+	97 (83–100)	54 (25–81)	83 (66–93)	88 (47–100)	2.09 (1.16–3.78)	0.06 (0.01–0.45)
HPV+ all types (MyHPV CHIP™) or LSIL+	89 (52–100)	58 (37–78)	44 (22–69)	93 (68–100)	2.13 (1.26–3.61)	0.19 (0.03–1.25)
HPV+, High-risk (MyHPV CHIP™) or LSIL+	89 (52–100)	62 (41–81)	47 (23–72)	94 (70–100)	2.37 (1.35–4.17)	0.18 (0.03–1.16)
HPV+, High-risk (BD Onclarity™) or LSIL+	97 (83–100)	54 (25–81)	83 (66–93)	88 (47–100)	2.09 (1.16–3.78)	0.06 (0.01–0.45)
HPV+ all types (MyHPV CHIP™) and ASCUS+	89 (52–100)	79 (58–93)	62 (32–86)	95 (75–100)	4.27 (1.89–9.62)	0.14 (0.02–0.90)
HPV+, High-risk (MyHPV CHIP™) and ASCUS+	67 (30–93)	92 (73–99)	75 (35–97)	88 (69–97)	8.00 (1.96–32.60)	0.36 (0.14–0.92)
HPV+, High-risk (BD Onclarity™) and ASCUS+	73 (54–88)	77 (46–95)	88 (69–97)	56 (31–78)	3.18 (1.15–8.77)	0.35 (0.18–0.67)
HPV+ all types (MyHPV CHIP™) and LSIL+	89 (52–100)	83 (63–95)	67 (35–90)	95 (76–100)	5.33 (2.12–13.44)	0.13 (0.02–0.85)
HPV+, High-risk (MyHPV CHIP™) and LSIL+	67 (30–93)	96 (79–100)	86 (42–100)	88 (70–98)	16.00 (2.22–115.14)	0.35 (0.14–0.88)
HPV+, High-risk (BD Onclarity™) and LSIL+	60 (41–77)	77 (46–95)	86 (64–97)	45 (24–68)	2.60 (0.92–7.32)	0.52 (0.31–0.88)

**Table 14 ijerph-17-01726-t014:** Performance of HPV test combinations and cytology for the detection of CIN2 or worse (Age ≥ 45).

Age ≥ 45	Sensitivity% (95% CI)	Specificity% (95% CI)	PPV% (95% CI)	NPV% (95% CI)	PLR (95% CI)	NLR (95% CI)
HPV+, all types (MyHPV CHIP™)	79 (58–93)	76 (64–85)	54 (37–71)	91 (80–97)	3.27 (2.03–5.24)	0.28 (0.12–0.61)
HPV+, High-risk (MyHPV CHIP™)	79 (58–93)	80 (69–89)	59 (41–76)	91 (81–97)	4.02 (2.37–6.82)	0.26 (0.12–0.57)
HPV+, High-risk (BD Onclarity™)	86 (74–94)	72 (60–82)	71 (59–81)	87 (75–94)	3.09 (2.10–4.56)	0.19 (0.10–0.38)
ASCUS or worse (ASCUS+)	83 (73–90)	62 (53–70)	56 (46–65)	86 (77–92)	2.15 (1.71–2.72)	0.28 (0.17–0.46)
LSIL or worse (LSIL+)	72 (60–81)	74 (66–81)	62 (51–72)	82 (74–88)	2.74 (2.01–3.75)	0.38 (0.27–0.55)
HPV+ all types (MyHPV CHIP™) or ASCUS+	92 (73–99)	50 (37–63)	40 (27–54)	94 (81–99)	1.83 (1.40–2.40)	0.17 (0.04–0.64)
HPV+, High-risk (MyHPV CHIP™) or ASCUS+	92 (73–99)	50 (37–63)	40 (27–54)	94 (81–99)	1.83 (1.40–2.40)	0.17 (0.04–0.64)
HPV+, High-risk (BD Onclarity™) or ASCUS+	96 (88–100)	56 (43–67)	63 (52–73)	95 (84–99)	2.17 (1.67–2.82)	0.06 (0.02–0.25)
HPV+ all types (MyHPV CHIP™) or LSIL+	92 (73–99)	59 (46–71)	45 (31–60)	95 (83–99)	2.24 (1.64–3.07)	0.14 (0.04–0.54)
HPV+, High-risk (MyHPV CHIP™) or LSIL+	92 (73–99)	59 (46–71)	45 (31–60)	95 (83–99)	2.24 (1.64–3.07)	0.14 (0.04–0.54)
HPV+, High-risk (BD Onclarity™) or LSIL+	96 (88–100)	62 (50–74)	67 (56–77)	96 (85–99)	2.57 (1.90–3.48)	0.06 (0.01–0.22)
HPV+ all types (MyHPV CHIP™) and ASCUS+	62 (41–81)	83 (72–91)	58 (37–77)	86 (75–93)	3.75 (2.01–6.99)	0.45 (0.27–0.76)
HPV+, High-risk (MyHPV CHIP™) and ASCUS+	62 (41–81)	88 (78–95)	65 (43–84)	87 (76–94)	5.16 (2.51–10.59)	0.43 (0.25–0.72)
HPV+, High-risk (BD Onclarity™) and ASCUS+	75 (62–86)	82 (71–90)	77 (64–87)	81 (70–89)	4.18 (2.50–6.98)	0.30 (0.19–0.48)
HPV+ all types (MyHPV CHIP™) and LSIL+	58 (37–78)	88 (78–95)	64 (41–83)	85 (75–93)	4.81 (2.31–10.01)	0.47 (0.29–0.77)
HPV+, High-risk(MyHPV CHIP™) and LSIL+	58 (37–78)	92 (83–97)	74 (49–91)	86 (76–93)	7.70 (3.11–19.09)	0.45 (0.28–0.73)
HPV+, High-risk (BD Onclarity™) and LSIL+	61 (48–74)	86 (76–93)	78 (63–89)	74 (63–83)	4.42 (2.40–8.14)	0.45 (0.32–0.63)

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
