# Peer review of "Utility of Human Papillomavirus Testing for Cervical Cancer Screening in Korea"

_ijerph, 2020, doi:10.3390/ijerph17051726_

Round 1

Reviewer 1 Report

Comments to Authors 

            The current study has showed that HPV testing showed higher sensitivity than cytology but the sensitivity and specificity of HPV testing had variation depending on the HPV testing method.

           Authors are kindly requested to emphasize the current concepts about these issues in the context of recent knowledge and the available literature. This articles should be quoted in the References list.

References

  1. Evaluation of satisfaction with three different cervical cancer screening modalities: clinician-collected Pap test vs. HPV test by self-sampling vs. HPV test by urine sampling. J Gynecol Oncol. 2019 Sep; 30 (5): e76. doi: 10.3802/jgo.2019.30.e76.
  2. High-risk human papillomavirus testing as a primary screening for cervical cancer: position statement by the Korean Society of Obstetrics and Gynecology and the Korean Society of Gynecologic Oncology. J Gynecol Oncol. 2020 Jan; 31 (1): e31. doi: 10.3802/jgo.2020.31.e31.
  3. Comparison of papanicolaou smear and human papillomavirus (HPV) test as cervical screening tools: can we rely on HPV test alone as a screening method? An 11-year retrospective experience at a single institution. J Pathol Transl Med. 2020 Jan; 54 (1): 112-118. doi: 10.4132/jptm.2019.11.29.

Reviewer 2 Report

*General Comments

 There are too many tables. Thus, readers cannot recognize these contens smoothly.

 The authors should arrange them.

*Special Comments

1) The authors should write pathological diagnosis in Material and Methods. Did they compare the screening methods

    (HPV or cytology) with pathological diagnosis?  Thus, they should write it in abstract, too.

2) There are too many tables in text. It is hard for us to understand them. The authors should reduce them.

   In particlular, it is hard for us to recognize the contents of table 12 and 13. The authors should re-write table 12 and 13.

3) In table 15, odds ratio is 19.19 in HPV(-) vs HPV(+) in normal cytology. On the other hand, odds ration is 7.9 in HPV(-)

   vs HPV (+) in abnormal cytology. Are they reverse?

Reviewer 3 Report

The study “Utility of Human Papillomavirus Testing for Cervical Cancer Screening in Korea” is a very important public health topic. The study has three strengths.

  1. The large sample size allows for subgroup analyses and it is a plus to this study.
  2. The authors conducted several robust statistical analyses in the study.
  3. The findings of the study are important to contributions to public health

Weakness

The whole paper needs to be rewritten. All most every sentence has some issues with them. Some are simple grammatical errors and others are sentence structures.

Examples of simple grammatical errors include

  1. The authors stated, “This study were performed to discover the utility of HPV testing in screening of cervical lesion and to provide the HPV prevalence and genotype distribution among a single center of Korea.”

The sentence should read “ This study was performed to ….HPV testing in the screening of cervical lesion…”

  1. The authors stated, “Cervical cancer is one of common cancer in Korean women and 2,856 new cases were detected and 834 deaths were occurred in 2019 [1].”

The sentence should read “Cervical cancer is the most common or one of the common cancers in Korean women … and 834 deaths occurred or were reported… “

  1. The authors stated, “Since cervical screening by the Papanicolaou test, the incidence and mortality rate are decreasing year by year in Korea.”
  2. This sentence has both simple error and structural problems. The simple is that the sentence should read like “… the incidence and mortality rates are ….” The structural problem is “Since cervical screening by the Papanicolaou test …” I believe the authors are trying to say that “Since the introduction of Papanicolaou test for cervical screening, the incidence and mortality rates of cervical cancer are decreasing year by year in Korea.”

As I said, these are just a few examples, throughout the whole paper, those types of simple errors can be found in almost every sentence.

Examples of sentence structure problems include

  1. The authors stated, “In 1983 and 1984, HPV 16 and HPV18 were isolated from cervical cancer respectively [3]”

This sentence does not make any sense. I know HPV 16 and 18 are associated with cervical cancer why were they isolated from cervical cancer? They need to clarify.

  1. The authors “15141 women [aged 15-97, 43(median); 43.36(mean)] who underwent both HPV testing and cervical cytology were enrolled in this study from Health checkup center [n=11,142, aged 23-86, 42(median); 41.5(mean)] and gynecology department [n=3,999 aged 15-97, 47(median); 47.5 (mean)] of Samsung Changwon Hospital.”

First, it is incorrect to begin a sentence with a figure and second, the sentence needs to be rephrased because it is confusing.

  1. The authors stated, “The Papanicolaou cervical cytology had been diagnosed in routine practice at department of pathology, Samsung Changwon Hospital without any intervention.”

This sentence does not make sense because the Papanicolaou cervical cytology cannot be diagnosed, we use Papanicolaou cervical cytology to screen for HPV infection.

These are just a few examples. So I believe the authors should carefully go over the whole manuscript to make sure those problems are fixed.
